# Adiposity Metabolic Consequences for Adolescent Bone Health

**DOI:** 10.3390/nu14163260

**Published:** 2022-08-10

**Authors:** Kátia Gianlupi Lopes, Elisana Lima Rodrigues, Mariana Rodrigues da Silva Lopes, Valter Aragão do Nascimento, Arnildo Pott, Rita de Cássia Avellaneda Guimarães, Giovana Eliza Pegolo, Karine de Cássia Freitas

**Affiliations:** 1Post-Graduate Program in Health and Development in the Mid-West Region, Federal University of Mato Grosso do Sul, Campo Grande 79070-900, Brazil; 2Institute of Biosciences, Federal University of Mato Grosso do Sul-UFMS, Campo Grande 79079-900, Brazil; 3Faculty of Pharmaceutical Sciences, Food and Nutrition, Federal University of Mato Grosso do Sul, Campo Grande 79070-900, Brazil

**Keywords:** pediatric obesity, body composition, bone health

## Abstract

Infancy and adolescence are crucial periods for bone health, since they are characterized by intense physical growth and bone development. The unsatisfactory acquisition of bone mass in this phase has consequences in adult life and increases the risk of developing bone diseases at more advanced ages. Nutrient deficiencies, especially calcium and vitamin D, associated with a sedentary lifestyle; lack of sun exposure; and epigenetic aspects represent some of the main risk factors for poor bone quality. In addition, recent studies relate childhood obesity to impaired bone health; however, studies on the adiposity effects on bone health are scarce and inconclusive. Another gap concerns the implications of obesity on child sexual maturity, which can jeopardize their genetic potential bone mass and increase fracture risk. Therefore, we reviewed the analyzed factors related to bone health and their association with obesity and metabolic syndrome in adolescents. We concluded that obesity (specifically, accumulated visceral fat) harms bones in the infant–juvenile phase, thereby increasing osteopenia/osteoporosis in adults and the elderly. Thus, it becomes evident that forming and maintaining healthy eating habits is necessary during infancy and adolescence to reduce the risk of fractures caused by bone-metabolic diseases in adulthood and to promote healthy ageing.

## 1. Introduction

Obesity in infancy and adolescence has been increasing in recent years. Over the next decade, it is predicted that 254 million children and adolescents (5–19 years of age) will have obesity worldwide [1]. In the European countries evaluated in the fourth data-collection round of the Childhood Obesity Surveillance Initiative (COSI), the prevalence of overweight (including obese) children between 7 and 9 years of age was 29% for boys and 27% for girls [2].

Some studies indicate that pediatric obesity increased due to the COVID-19 pandemic. A cohort study on individuals between 5 and 17 [3] revealed the highest weight gain, corroborating the results of Lange et al. [4] for children and adolescents between 6 and 11, who had the highest gains in body mass index (BMI) compared with other age groups.

Thus, the metabolic disorders related to obesity most commonly identified in adults are already found in infancy and adolescence. Early identification of arterial hypertension, type 2 diabetes, non-alcoholic steatosis [5,6], and metabolic syndrome (MS) [7,8,9,10,11,12] is fundamental for the development of prevention strategies.

Factors such as prenatal and lactation nutrition, bottle feeding or breastfeeding, food with functional properties, and an adequate intake of omega-3 fatty acids, calcium, zinc, vitamins (A, C, D, and E) and folate contribute to preventing and treat childhood obesity [13].

Vitamin D deficiency affects over 1 billion people worldwide [14]. In Germany (96%) [15], Iran (95.6%) [16], and Canada (93%) [17], vitamin Ddeficiency in children and adolescents is over 90%. It is worth pointing out that the prevalence of hypovitaminosis D among adolescents is higher among females and associated with low serum calcium levels [18].

In this context, vitamin D and calcium deficiencies are frequent in several countries irrespective of nutritional status, but their magnitude is higher in overweight children [19]. Among young people, a dairy-poor diet is a primary cause of calcium deficiency [20,21]. Moreover, food that is a natural sources of vitamin D (e.g., oily fish, cod liver oil, mushrooms, and yeasts) is not regularly consumed by most people, especially adolescents. Fortified orange juice (100 IU) and cereals (40 IU) can provide a daily source of this vitamin [22,23], but food fortification is not standardized in all countries. Consequently, cutaneous synthesis represents the primary precursor of this vitamin through sun exposure. However, several factors can interfere with this process—such as latitude, season, air pollution, obesity [24,25], clothing, skin pigmentation, and sunblock use [26].

The observation that inadequate levels of the abovementioned nutrients are associated with the pathogenesis of chronic conditions—such as MS, type 2 diabetes, arterialhypertension, obesity, and cardiovascular diseases—has caused concern [27,28,29].

The combined deficiency of these nutrients deserves special consideration in children and adolescents [13] because vitamin D plays an essential role in regulating levels of calcium, phosphorous [30], iron, magnesium, and zinc [31]. It is also involved in insulin secretion, glucose homeostasis [31], and parathyroid hormone levels [32]. Calcium is an essential nutrient for maintainingbone health since it contributes to mineralization and skeleton hardness, preventing osteoporosis and fractures in adult life. Adequate ingestion of this nutrient is fundamental for developing and maintaining the peak of bone mass during adolescence [33,34].

Besides its musculoskeletal actions, vitamin D promotes several biological functions (e.g., action in the keratinocytes [35], antimetastatic and antitumorigenic activity in various cancer cells [36], immunomodulatory responses in macrophages and activated lymphocytes B and T [37], cardiovascular protection and prevention of pregnancy complications [38]).

Although vitamin D and calcium deficiency is commonly reported, little is known about the action of these nutrients in preventing or treating obesity. Therefore, this revision analyzes factors related to bone health and its association with obesity and MS in infants and adolescents.

## 2. Physical Growth and Pubertal Development

The acquisition of final adult height is mainly due to genetic factors (60–80%) [39]. Socioeconomic factors, parental education levels, diseases and nutrition influence approximately 20–40% of this growth [40], representing an intervention opportunity to ensure that a child reaches at least the parental-estimated mean height [41].

Puberty is an essential time of substantial bone growth and is therefore sensitive to external influences that have strong effects, such as diet, physical exercise, lifestyle, and medication [42]. It is also a period of biological maturity, marked by the appearance of secondary sexual characteristics and changes in body composition. This stage allows the evaluation of sexual maturity and its correlation with phenomena such as the age of menarche, growth spurt, and final height [43]. An increase in growth velocity occurs at puberty, which is known as the pubertal growth spurt. That acceleration occurs after a period of slow growth in infancy, followed by a phase of higher velocity and, finally, a reduced speed until adult height is attained. Puberty and its associated growth spurt are expressed earlier in girls than in boys; however, boys present a mean height gain superior to girls [44].

Carrascosa et al. [45] developed a longitudinal study with 1453 healthy children, classified according to the age at which the pubertal spurt began: very precocious(girls: 8–9; boys: 10–11); precocious (girls: 9–10; boys: 11–12); intermediate (girls: 10–11; boys: 12–13), late (girls: 11–12; boys: 13–14); and very late (girls: 12–13; boys: 14–15). The greatest differences in growth speed were observed between the groups classified as very early and very late (5.1 cm/year for girls 13–14 and 6 cm/year for boys 15–16). In girls, the growth speed peak was reached in the Tanner breast stages II–III. The mean age of menarche differed significantly between groups. Significant and clinically important differences regarding characteristics of pubertal growth were detected in the five groups of pubertal maturity, which indicated the relevance of these findings to improving the clinical evaluation of growth according to the specific development of each youngster.

In a cohort study, McCormack et al. [46] found that even after the growth-speed peak, significant bone gain occurred when adult height was attained. Thus, the final phase of adolescence represents an opportunity to intervene to improve the gain of bone mass.

Therefore, it is recommended that the evaluation of adolescent nutritional status be carried out considering the pubertal stage since chronological age may not represent a safe parameter. Changes in nutritional status can strongly influence sexual maturity, especially the distribution and deposition of body fat. Thus, excessive visceral adiposity is associated with anticipated menarche in girls and precocious or late puberty in boys [47].

### 2.1. Nutritional Aspects for Growth Optimization

Nutrition and growth are closely associated because children do not reach their genetic height potential if their basic nutritional needs are not met [48].

To support this intense physical growth, an adequate supply of energy is necessary, of which 4% is destined for physical growth [49]. Consumption of proteins is also increased during this period to increase the supply of essential amino acids crucial for the puberty growth spurt. The recommended intake is defined by the relation of grams of proteins to body weight, which for adolescents is 1 g/kg of body weight [50].

A cohort study conducted by Chevalley et al. [51] of adolescent pre-pubescentboys until young adulthood—totaling 15 years—identified a positive synergy between adequate protein ingestion and the physical activity on bone resistance and mass. That association of factors showed an incremental increase in the size and width of the femoral head; moreover, the ingestion of proteins added bone growth in both transversal and longitudinal dimensions, probably influenced by somatomedine C. Such findings show the importance of both nutritional and physical activity interventions in the pre-pubescentperiod for attaining higher peak bone mass and so modify the risk of developing fractures and bone fragility.

Calcium represents another essential nutrient for the maintenance of bone health since it contributes to mineralization and skeleton hardness [33,34] in addition to being involved in blood coagulation, muscle contraction, and transmission of nerve impulses [52]. Mineralized tissue contains 99% of total body calcium and it is suggested that mineralization occurs inside matrix vesicles, where chondrocytes and small, extracellular vesicles originating from osteoblasts initiate the process. Inorganic phosphate is transported into these vesicles through paths dependent and independent of sodium. The bond between calcium and inorganic phosphate forms hydroxyapatite crystals that migrate to the collagen fibrils to promote mineralization of the extracellular matrix [53,54].

Apart from calcium, other minerals and elements are involved in bone growth, some as bone mass constituents (magnesium and fluorine), and components of the enzymatic system involved in matrix mechanisms, such as zinc, copper, and manganese. A diet deficient in these nutrients reduces bone growth during its definitive formation. Vitamins D, C, and K also play essential roles in calcium metabolism by acting as cofactors of key enzymes in bone metabolism [55]. The need for calcium varies with age range since children and adolescents between 9 and 18 have a minimal nutritional recommendation of 1300 mg/day, according to the recommended dietary allowance (RDA) but should not exceed the maximum tolerable ingestion level of 3000 mg/day [56].

The bone mineral density (BMD) of adults depends on the peak bone mass acquired by the end of the second decade of life. Despite a lack of consensus about the age when peak bone mass occurs, it is recognized that about 40% is accumulated between 11 and 14 in girls and 13 and 17 in boys [52]. Pre-pubescentchildren between 3 and 10 years retain approximately 120 mg/day of calcium for skeleton growth, and this demand increases to over 600 mg/day at puberty [57].

Concerning stature gain, some studies indicate that excess weight (overweight, obesity) can have negative consequences on stature. Pinhas-Hamiel et al. [40] assessed the impact of body weight on the stature of adolescent army recruits in Israel and found that those overweight and obese had a higher risk of staying below their genetic height potential than those of normal weight, whereas women with overweight/obesity had a 73% increased risk of low stature; in contrast, girls of low weight became taller than predicted and had a double probability of growing tall. This influence was less significant in boys, indicating a sex-related difference in weight influencing height. Holmgren et al. [58] also pointed out the relation between obesity in infancy, specifically severe obesity, and the harm to height gain in the pubertal growth spurt compared to those with less severe obesity. Their findings emphasized the importance of infant BMI as a modifying factor of pubertal growth under different conditions of nutritional status.

Obesity has been associated with hyperandrogenism in girls, but the associated mechanisms are still not fully clarified; some studies point out that individuals with hyperandrogenism have metabolic and neuroendocrine alterations similar to characteristics associated with polycystic ovaries syndrome (PCOS) in adults. One of the proposed mechanisms is that peri-pubescentgirls with high levels of gonadotrophin-releasing hormone (GnRH) for any cause could develop a lower susceptibility to this hormone, producing more of the luteinizing hormone (LH) than the follicle-stimulating hormone (FSH). Low FSH production hinders follicular development and ovulation, yet LH increases production of ovarian androgens, potentially leading to hyperandrogenism [59]. Increased levels of androgens in pre-pubescentgirls can facilitate the increase in GnRH secretion that could be related to earlier pubertal onset [60].

Obese girls can also have hyperandrogenism due to increased production of total testosterone and reduced sexual hormone binding globulin (SHBG), which represents a risk factor for the development of SOP [47]. A low level of SHBG has been identified as a strong predictor of insulin resistance and SM risk; however, this causal relation occurs only before menarche and not after it, and more studies are needed to elucidate the mechanisms involved in cardiovascular risk in peripubertal girls [61].

Infant obesity is associated with the acceleration of linear growth during pre-puberty, which is possibly due to early estrogenization and action of insulin-like growth factor1 (IGF-1), as well as the negative impacts on bone mass and bone mineral density in both sexes [47].

Vitamin D is an essential pro-hormone for normal growth and development [62] and is relevant for the formation of bones and teeth because it is responsible for fixing calcium and phosphorous [63]. Furthermore, it has a function in immunity, reproduction, and insulin secretion. With parathormone, vitamin D mobilizes calcium from the bones and increases tubular renal reabsorption of calcium and phosphorous [32].

Vitamin D deficiency can have severe clinical consequences such as hypocalcemia, which leads to rickets and osteomalacia or even death. Rickets result from defective bone mineralization during skeleton growth in children [63]. Osteomalacia in both children and adults results from the bone mineralization failure of the preformed osteoid tissue produced by the osteoblasts. The cause is calcium and phosphorous deficiency, which may be secondary to the lack of vitamin D (calcitriol), which is needed for intestinal calcium absorption; thus, osteomalacia in children coexists with rickets [64].

The amount of dietary vitamin D required depends on factors such as geographical location and the amount of sun exposure since this nutrient also comes from the reaction of sunlight with subcutaneous tissue. Thus, to stimulate and increase the vitamin D production by the organism, it is important to sunbathe and practice open-air physical activities [65].

### 2.2. Nutrients and Bone Mineralization (Impact of Nutrients on Bone Health)

The importance of sufficient extracellular levels of ionic calcium (Ca^2+^) and inorganic phosphate (P_i_) for adequate bone mineralization is well-known; nevertheless, experimental studies found that low levels of inorganic phosphate are more harmful for mineralization than ionic calcium deficiency causing osteomalacia, characterized by a non-mineralized bone matrix [66].

Another nutrient that has an impact on bone health, though a negative one, is arachidonic acid (AA), a long-chain polyunsaturated fatty acid. In an experimental study on the effect of AA on bone mass, quality, and adiposity in animals, growing male rats fed a hyperlipidic diet demonstrated that added AA enhanced obesity effects, resulting in a higher percentage of body fat (12%), higher body weight (6%) and a higher concentrations of leptin (125%). AA also led to the reduction inbone minerals in some bone parts, but did not affect bone resistance in femoral diaphysis [67].

Another type of fatty acid harmful to bone health is saturated fat, which Corwin et al. [68] found to be negatively associated with the bone mineral density (BMD) of the hip bone. Diets rich in industrialized food also hinder bone health since they frequently contain high levels of saturated fat. Thus, the effects of dietary fat on bone healthdepend on the quantity and type of the predominant fatty acid since fat is a fundamental nutrient for the absorption of vitamin D, which is liposoluble. Therefore, the effect of fatty acids on bone metabolism and general health differ [69].

Polyunsaturated fatty acids (PUFAs), especially omega-3 fatty acids, have positive effects on bone mass and quality, which are possibly attributable to reduced production of prostaglandins E2 (PGE 2) and inhibition of the receptor of the differentiatedosteoclasts caused by the ligand NF-κβ [70,71]. These fatty acids promote bone formation, thereby increasing the differentiation and survival of osteoblasts [72] and their association with ingested proteins, supplementaryvitamin D, high levels of growth hormone (GH), and fibroblast factor 23 (FGF23). In contrast, a high saturated fat intake, low calcium ingestion, hyperparathyroidism (HPT), and high oxidative stress favor osteoblastogenesis and bone reabsorption [69].

### 2.3. Calcium Ingestion by Adolescents

One factor that interferes with the peak development of bone mass in adolescents and the preservation of this mass in adults is calcium. This mineral is responsible for 30–35% of bone mass and is a crucial part of its strength. Low ingestion by some populations can cause complications in the peak development of bone mass in adolescents, mainly girls, thereby increasing the risk of osteoporosis and fractures in later life [73,74].

Studies in different countries showed that, in the growth period and puberty, the consumption of dietary calcium was below the 1300 mg/day recommendation proposed by the dietary reference intakes (DRIs) [56]. In Australia, adolescents had a mean daily consumption of calcium between 583 mg (girls) and 738 mg (boys) [75]. American adolescents had mean ingestion of 752 mg calcium [76]; and adolescents in Estonia, 786 mg [77].

In Spain, the “Healthy Lifestyle in Europe by Nutrition in Adolescence Cross-Sectional Study” (HELENA-CSS) found that about 96% of adolescents did not reach the recommended daily calcium intake, and that about 50% ingested one or less than one portion of milk, which was more critical for girls. The results showed a positive relation between milk ingestion and bone mineral content (BMC) and BMD but only in boys. In contrast, a small but crucial positive relation for girls was detected between serum levels of 25-hydroxyvitamin D (25(OH)D) and some bone sites [78].

In the city of Campinas (SP), Brazil, transversal population base research confirmed that the lowest dietary levels of calcium were in the diet of adolescent girls in the lower socioeconomic strata and by individuals with other unhealthy behavior: consumption of alcohol and tobacco but low consumption of fruit [20]. Peters et al. [79] found 682 mg/day and 124 UI/day for calcium and vitamin D consumption, respectively, with no significant difference between the sexes. This suggests the need to intensify education about the importance of these nutrients since both calcium and vitamin D are related to bone health and to consider food fortification with these nutrients to ensure their adequate intake. In contrast, a bad diet, characterized by highly processed foods, was related to the lowest levels of calcium ingestion by adolescents in addition to being the main contributor to excessive energy intake and weight gain [75].

Lappe et al. [76] investigated a possible relation between increased dairy calcium ingestion and reduction in body fat among female adolescents who habitually had a low calcium intake compared with the control group, which maintained its usual diet. The intervention group gained body fat in the same proportion as the control; hence, it was not an effective strategy to reduce fat mass. Nevertheless, an adequate calcium intake in girls is needed due to the bone build-up at the onset of adolescence. A clinical trial on the effects of consuming dairy products on bone mass gain by adolescents with overweight and eutrophy showed that >3 portions of dairy products/day produced a significantly higher gain in the tibial region, without a difference in bone gain in other body regions compared to twoportions of dairy products/day [80].

In contrast, a multiple regression analysis between consumption of dairy products and anthropometric health indicators in adolescents showed a significantly positive association with fat-free mass in boys but not in girls. Thus, the role of dairy products in the trend for central adiposity and body composition seems to be gender specific [81].

In a systematic review of controlled trials concerning the ingestion of dairy products and the linear growth and bone mineral content in infancy and adolescence, De Lamas et al. [82] found that six of seven articles showed a positive relation with bone mineral content. Those results were verified in various body regions, but are considered short-term studies (between 14 weeks and 2 years); thus, long-term intervention and cohort studies starting at infancy are necessary.

In Spain, Marcos-Pasero et al. [83] detected lower levels of calcium and dairy product ingestion in overweight and obese school children aged 6–9 and a significant inverse relation between BMI and the quantity of calcium/day. Blood pressure was inversely related with daily calcium and its recommended nutritional percentage. Furthermore, they emphasized preventive tools to stimulate calcium intake to meet the daily needs. During the school phase, milk and dairy products complete the diet and fulfil calcium needs. Thus, the eating pattern has a fundamental role in childhood health, which is reflected in body weight and blood pressure.

A study by Brazilian scholars verified an inverse relation between calcium ingestion and abdominal adiposity and subclinical inflammation levels. Almost the entire study sample—97%—had inadequate ingestion. Those results indicated that insufficient calcium represented a cardiometabolic risk factor even in infancy [84]. Thus, it became evident that maintaining healthy eating habits during infancy and adolescence affects adult life and healthy ageing. A Korean study verified the combined effect of calcium intake and physical activity on BMD in adolescents. Those who did not drink milk or engage inphysical activity were less prone to higher BMD than those who drank milk and had high-level physical activity, thus indicating a synergy between milk ingestion and moderate-to-vigorous physical activity and bone health in adolescents [85].

## 3. Nutritional Status of Vitamin D Insufficiency or Deficiency

The DRI for vitamin D, established from a review of the existing evidence, considers skeleton and non-skeleton effects and its role in calcium absorption and prevention of rickets and or osteomalacia. In addition, sun exposure in winter was a minimal contributing factor to vitamin D levels. Thus, the ingestion of 600 UI/day of vitamin D is recommended for youngsters between 10 and 19 [56]. However, most of the global population, including adolescents, did not reach this recommended amount through food alone.

Julian et al. [86] found a positive association between ingestion of calcium and vitamin D with serum levels of vitamin D in adolescents from several European countries at various latitudes. An increase of 10 mg of calcium/day increased vitamin D serum concentration from 12 to 25 nmol/L (OH). However, such association occurred only in adolescents in Central Europe, possibly from a higher dependence on food calcium because of limited sun exposure. They also point out the need for longitudinal studies to verify this possible association because their study was transversal, which did not allow the establishment of a cause–effect relation.

Blood concentrations of vitamin D dependent on the capacity of cutaneous synthesis can vary according to skin type, geographic area, season, and adequate plasmatic levels. Nutritional recommendations in the guides of several countries vary. For example, Australia, New Zealand [87], India [88], and the U.S. [89] accept values above 50 nmol/L. In contrast, Italy [90], the European Society of Pediatric Nephrology [91] and the Society for Adolescent Health and Medicine [92] consider plasmatic levels > 75 nmol/L to be sufficient.

The Endocrine Society of Clinical Practice Guideline (2011) recommends evaluating the vitamin D status in individuals at risk of hypovitaminosis D by measuring the circulating serum level of 25 (OH)D. Sufficiency of vitamin D is considered to be 25 (OH)D when PTH levels are reduced until they stabilize. The literature considers indices above 30 ng/mL. The diagnosis of deficiency/insufficiency is given when an individual has levels < 30 ng/mL because reaching sufficiency status requires an increase in PTH correlated with the reduction in vitamin D serum levels; thus, deficiency is defined by the level of 25 (OH) D below 20 ng/mL and insufficiency between 21 and 29 ng/mL [89,93].

Frequently, deficiency of vitamin D is asymptomatic; however, records of symptoms such as unspecific bone pain (children can feel pain in the lower limbs), low tolerance to exercise, fatigue, and muscle ache. Children and adolescents with a deficiency can present symptoms of hypocalcemia, cramps, and muscular weaknesses or corpopedal spasms [87].

The Statement of Society for Adolescent Health and Medicine [92] reports that a daily supplement of 600 IU of vitamin D for healthy adolescents can reduce deficiency because the mean ingestion of this micronutrient is below the recommended level. However, a randomized clinical trial to evaluate the efficacy of different doses of vitamin D serum concentrations of 25 (OH)D and other parameters in overweight and obese children and adolescents concluded that a higher dose (2000 UI/day) was more effective in raising 25 (OH)D levels in obese individuals [94].

Furthermore, a systematic review with a meta-analysis comparing obese and eutrophic adolescents identified a higher relative risk of hypovitaminosis D in the obese adolescents, reinforcing the importance of a healthier lifestyle and evaluating levels of 25 (OH) D in obese children and adolescents [95]. Such investigation allows the early identification and intervention of individuals more prone to hypovitaminosis D and the investigation of its possible causes.

Since the diet can only provide small quantities of vitamin D, a more effective strategy to increase its level is through greater consumption of enriched foods, such as dairy products and breakfast cereals. The increased prevalence of its deficiency in India and lack of orientation on adequate supplementation led to the creation of recommendations and treatments. The focus was directed toward long-term public policies to fortify staples and other daily consumed foods with vitamin D and calcium to maintain serum levels [88]. However, it is known that one of the industrial challenges in the fortification of drinks and food products is the solubility of this micronutrient in fat [96].

The Study of Cardiovascular Risks in Adolescents (ERICA) in Brazil determined that 63% of adolescents had vitamin D serum levels below 30 ng/mL irrespective of region. Girls, non-whites, and private-school students were the most affected because of their diet; obese boys were also at a high risk for developing hypovitaminosis D. A hypothesis for the higher vitamin D deficiency in girls is that they engage inless physical activity outdoors compared to boys [97], are three times more likely to use sun cream, and spend less time in the sun [98]. Therefore, lifestyle changes are needed along with better food choices and more physical activity in the open [99].

Low vitamin D levels in otherwise healthy adolescents raises concerns about intense bone growth in the critical late teens. Thus, adequate circulating concentrations of 25 (OH)D from supplementation when needed is associated with other aspects of a healthier lifestyle. Standardization of adequate levels for this age range should take into account peculiarities related to sex and ethnicity [52,100].

Despite the lack of a direct association between the inflammatory process and levels of liposoluble vitamins, the literature reports that factors such as obesity and inflammation should be considered when assessing biochemical dosages to avoid misinterpretation of laboratory tests. Some studies indicate that pro-inflammatory cytokines lower the hepatic production of the carrier of these vitamins, as well as increasing capillary permeability and proportionate micronutrient sequestration to other organs, including the liver. A study in Recife, Brazil, on adolescents of both sexes aged 12 to 19 showed an inflammation risk due to the build-up of abdominal fat. Regarding general and abdominal adiposity, an analyses of serum concentrations of liposoluble vitamins and nutritional status revealed that boys were at a higher risk of low blood concentrations of 25 (OH)D. The conclusion is that abdominal adiposityincreases inflammation risk and that plasmatic alterations of liposoluble vitamins differ between the sexes [101].

An assessment of the variation in vitamin D levels in the U.S. showed that the prevalence of insufficiency declined from 21% to 17.7% from 2011 to 2014. The reduction could be related to people at risk of deficiency and inadequacytaking supplements and enriched drinks and food. Such results demonstrated that strategies of food supplementation can be beneficial for reducing hypovitaminosis D [102].

Another observation by Rodrigues et al. [103] was that the unsatisfactory food profiles of Brazilian adolescents were mainly the result of skipping breakfast and dinner. Missing meals can be associated with a poor diet because of a low consumption of fruit, vegetables, and dairy and a high ingestion of fat and sodium. As already stressed by other authors, adolescents who have breakfast daily have a higher ingestion of fruit, milk, and other dairy products and lower ingestion of sodium––food habits considered favorable for bone health mainly because of calcium, vitamin D, and fiber [79].

Concerning life habits, Voráčováet al. [104] identified a positive association between active leisure activity and healthy eating habits of Czech adolescentsbetween 11 and 15, who frequently consume breakfast, fruit, and vegetables, and consume fewer sugary drinks and salty snacks, especially in front of the television or computer. The results indicated that poor food habitsand a lack of active leisure activity represent risk factors for adolescents.

## 4. Infant–Juvenile Obesity

Worldwide, infant–juvenile obesity represents one of the main problems of public health, independent of the level of development or income [105].A systematic analysis of the Burden of Disease in 195 countries for the prevalence of overweight and obesity in children, adolescents, and adults from 1980 to 2015 identified 107.7 million obese children and adolescents in 2015. In 70 countries, the number more than doubled in 35 years [106]. Over the last 40 years, the number of obese children and adolescents increased more than 10 times—from 11 million to 124 million—according to 2016 estimates [105]. Another alarming revelation is that this condition affected children even younger than five, representing over 38 million [107].

Child obesity affects several organ systems—such as the endocrine, gastrointestinal, pulmonary, cardiovascular, and musculoskeletal—resulting in an increased risk of developing hyperinsulinemia, insulin resistance, pre-diabetes, and type 2 diabetes [108,109,110]. In addition to dyslipidemia, obstructive sleep apnea and hepatic steatosis [109,110]. Any of these changes can further damage the health and quality of life of those children and adolescents.

According to Skinner et al. [109], children and young adults classified as severely obese have a higher prevalence of cardiometabolic risk factors, a major public health concern as cardiovascular diseases continue to stand out as one of the leading causes of death along with arterial hypertension, dyslipidemia, and hyperglycemia [111,112].

The International Childhood Cohort Consortium, a cohort study by Koskinen et al. [113], showed that the concomitance among obesity, hypertension, and dyslipidemia in adolescence are increased predictors for developing carotid intima media thickness, which represents a higher risk of cardiovascular occurrences. They also concluded that obesity was the factor most strongly associated with that alteration, increasing the risk by 3.7 times.

In Denmark, Bjerregaard et al. [108] found that the risk of overweight and obese 7-year boys developing type 2 diabetes can be minimized by maintaining a healthy body weight at puberty up to adulthood. Obesity at 7 or excess weight at 13 (pubertal phase) have partially reversible effects. Thus, the earlier the onset of excess weight, the higher the risk for type 2 diabetes, compared to development in adulthood. In contrast, the cohort study by Fan, Zhu, and Zhang [114] demonstrated that excess weight in infancy increases cardiometabolic risks, though such effects can be reduced if bodyweight returns to normal in adulthood.

Geserick et al. [115] brought a very important contribution to the dynamics of the beginning of obesity and the annual increase in BMI in children and adolescents in Germany. Prospective and retrospective analyses were made on the course of the BMI in a sample of 34,196 individuals from infancy to adolescence. A retrospective analysis determined that approximately half of the obese adolescents had a history of overweight or obesity from age 5 onwards. In the prospective analysis, it was found that about 90% of the children who were obese at 3 years of age had overweight or obesity in adolescence. In adolescents that were obese, the highest elevation in BMI occurred between 2 and 6 years of age. Besides, the rate of overweight or obesity in adolescence was higher in children born with high weight for gestational age.

Zou et al. [116] reported a positive association between high weight at birth (>4 kg) and infancy and excess weight (overweight or obesity) between 6 and 18 in China. Thus, weight control at birth—including monitoring the gain of gestational weight—can play a crucial role in the prevention and control of infant–juvenile overweight or obesity.

### 4.1. Relation between MS and Vitamin D Deficiency in Adolescents

Vitamin D deficiency is highly prevalent worldwide and has been studied mainly for its deleterious effects on bone health. There is also evidence indicating extraskeletal effects and concomitance between MS and vitamin D deficiency; moreover, the supplementation of this vitamin can improve the metabolic parameters of MS [117].

However, the interrelation between deficiency of vitamin D and MS is still not clear [118,119]. It is known that vitamin D plays a fundamental role in the regulation of glycemia by stimulating insulin secretion from β-pancreatic cells [120,121]. Vitamin D receptors in several tissues and organs can be a probable explanation for these findings [122].

It should be noted that there is a wide discussion whether this vitamin should be defined as a hormone rather than a vitamin, considering that it has the ability to be synthesized endogenously, unlike other vitamins that need to be obtained exclusively from food or supplements. Furthermore, vitamin D can be synthesized by cells and triggers different effects in other organs or target cells (which do not correspond to its place of origin) [123]. Therefore, evidence prevails for its definition more as a hormone or as a prohormone than as a vitamin [124].

Some studies investigate the relationship between a deficiency of vitamin D and MS. Kim, Hwang, and Song [9] found that 78% of Korean adolescents had vitamin D deficiency but only associated with one of the components of MS, causing an increased risk of fastening glycemia to be 207 times higher in individuals with hypovitaminosis D. In China, Fu et al. [10] developed a cohort study of people between 14 and 28 who had a high risk of MS. They demonstrated that the levels of 25 (OH)D were significantly lower in those who had a diagnosis of MS, obesity, elevated triglycerides, and type 2 diabetes. They also detected a negative correlation between vitamin D levels and neck circumference, percentage of body fat, cholesterol LDL, fastening glycemia, and glycemia values obtained from a glycose tolerance test after 2 h.

In the U.S., a transversal study evaluated the association between serum levels of 25 (OH)D and metabolic parameters of adiposity, serum lipids, fasting glucose, and insulin resistance in children and adolescents from 6 to 18; only a high risk of obesity, low levels of HDL-c and insulin resistance were associated with vitamin D deficiency, especially in girls [11].

Gannagé-Yared, Sabbagh, and Chédid [125] evaluated the relation between vitamin D levels and the lipid profile in Lebanese schoolchildren of different socioeconomic levels and concluded that 25 (OH)D levels were independent and inversely correlated with non-HDL cholesterol and triglycerides but positively correlated with cholesterol-HDL.

In Brazil, some studies were performed with this focus. Filgueiras et al. [126] evaluated the association between vitamin D intake with dyslipidemia and an insufficiency/deficiency of vitamin D in the 8–9 age range and concluded that the high prevalence of inadequate ingestion of this vitamin was associated with low levels of HDL-cholesterol, and 56.2% of vitamin D insufficient/deficient levels. That result was similar to the study by Queiroz et al. [127], who identified a vitamin D insufficiency/deficiency prevalence of 57.3% in Brazilian adolescents aged 15 to 19; however, an inverse association of vitamin D levels with cardiometabolic markers was detected, especially triglyceride levels, which were independently associated with sex.

Clinical trials using several methods of vitamin D supplementation evaluated its effects on MS parameters. In Iran, Kelishadi et al. [128] applied 300,000 UI of vitamin D weekly for 12 weeks and observed favorable effects on insulin resistance, MS, and triglycerides in obese children and adolescents. In contrast, Sethuraman et al. [129] supplemented only 50,000 UI of vitamin D weekly for 12 weeks in obese Afro-American adolescents and showed that improved vitamin D levels had a positive correlation among 25 (OH)D serum levels, fasting insulin, and HDL levels thereafter. A randomized clinical and triple-blind study verified that obese adolescents deficient in vitamin D who were taking 50,000 UI of vitamin D weekly, following a healthier diet and doing physical activity improved body composition (body mass and BMI), and the action of insulin and other metabolic parameters [130]. Rajakumar et al. [131] conducted a randomized clinical trial on overweight adolescents who were supplemented daily with 1000 and 2000 UI compared with 600 UI of vitamin D, identified improvements in blood pressure, fasting glycemia, and sensibility to insulin.

Xiao et al. [12] found that children with inadequate vitamin D levels had a higher cardiovascular risk; moreover, a higher risk of hypertension and high total cholesterol levels occurred in vitamin D-deficient girls, but not boys. According to those authors, this difference can be attributed to girls having lower vitamin D ingestion and a lack of open-air activity. They suggested that sun exposure and adequate supplementation should be encouraged to prevent cardiometabolic risks, especially in children with obesity.

### 4.2. Relation between Obesity and Bone Health

Adipose tissue plays a vital role in bone metabolism, mainly due to the production of adipokines, some of which act positively in bone formation [132]. However, obesity can also increase bone reabsorption through the increased release of pro-inflammatory cytokines (Figure 1), such as tumor necrosis factor-α (TNF-α) and interleukin 6 (IL-6), which stimulate the formation and activity of osteoclasts through the receptor activator of the nuclear factor bappa-beta ligand (RANKL)/(RANK)/Osteoprotegerin (OPG) pathway [133,134,135].

Another point is that, despite fractures of extremities being frequent in infancy and adolescence, there is evidence that excess body fat increases fracture risk. Kessler et al. [136] observed that the risk increased with body weight as children and adolescents with severe obesity had an increased fracture risk of around 50% in the foot, ankle, knee, and leg. In contrast, data suggest a higher CMO in obese children than in those with normal weight. However, the relationship between adipose tissue and bone can be explained by the fact that both adipocytes and osteoblasts originate from the same cells, the mesenchymal multipotent stem cells (MMSC). In obese individuals, differentiation of MMSC into adipocytes overrides the formation of osteoblasts, which can compromise bone quality [137].

Although obesity is still not considered a direct cause of osteoporosis, there are signs of that correlation. One mechanism that can explain such an influence is how obesity stimulates pre-osteoblasts to differentiate into adipocytes instead of osteoblasts, which increasing bone fragility by filling the cavities of bone marrow with adipocytes instead of bone trabecula [133].

In the specific literature, there are still disagreements about the real consequences of adiposity on bone mass. Some point to a negative impact [138,139,140,141], mainly concerning body fat mass, while others indicate positive [142,143,144,145] or neutral effects [146].

Mosca et al. [138] evaluated the impact of excess body fat on bone remodeling in adolescents distributed according to the classification of nutritional status into normal weight, overweight, obese, and extremely obese. They found that the highest fat mass and percentage of body fat in girls and the low levels of bone-remodeling biomarkers demonstrated the negative consequences of excess body fat on bone health.

Gállego-Suárez et al. [139] also identified a reduced BMD area of the total body and lumbar spine, with body percentage and bone mass for both sexes and all races. The negative effects of obesity were less expressive in the pelvic region, which was expected because this body region supports a higher weight load.

A cohort study in Estoniaof overweight and obese boys between 10 and 11 in the pubertal phase assessed the impact of extensive BMI gain over three years on bone mineral characteristics. The conclusion was that excessive mass gains of total body fat and the percentage of body fat could explain the lower increments in bone mass in adolescents who gained more weight during puberty [147].

Using data from a Boston cohort of children aged between 6 and 10, Rokoff et al. [140] evaluated the associations of total body mass (fat-free mass + total fat mass), components of total body mass (fat-free mass and total fat mass), and components of total fat mass (truncal and non-truncal) with the BMD area. The researchers determined that children with a higher total body mass and fat-free mass had higher Z-scores. The association between the highest levels of fat mass and the lowest Z-score levels for the BMD area was related to the pattern of fat deposit, especially abdominal adiposity. Those findings corroborated the supposition that the effect of adipose tissue on the bone can be related to its distribution, which is mainly attributed to visceral fat, which has an origin and function different from that of subcutaneous adipose tissue cells. In addition, it secretes pro-inflammatory cytokines—such as TNF-α and IL-6—which induce bone reabsorption and hinder bone development as mentioned above [133,148].

Liang et al. [141] evaluated the association of body fat and its distribution with BMD in Chinese children between 6 and 10 and concluded that body fat has a negative effect on BMD, mainly in children who have a distribution pattern of android fat, possibly for the higher build-up of visceral adipose tissue. On the other hand, Streeter et al. [149] performed a longitudinal study over seven years with a cohort of 307 children and adolescents aged 9 to 16 to evaluate the effect of body fat on bone growth over time. They verified that the highest body fat values were associated with thicker and denser bones in both boys and girls, concluding that body fat did not seem to harm bone quality. Nonetheless, the sample involved only Caucasians, and all tested individuals were healthy, without any consequences from metabolic disorders for bone health that are mainly related to obesity.

Jeddi et al. [142] and Kim et al. [143] found through transversal studies that excess weight (overweight/obesity) had a positive effect on BMD; however, lean mass was considered one of the most important predictors of this effect. Soininen et al. [144] obtained similar results as both muscle mass and fat mass were strongly associated with BMD in normal-weight pre-pubertal boys and girls. Kouda et al. [145], from a prospective cohort study, showed that the fat mass was positively associated with bone mass in pubertal children but only for those with a low or normal lean soft tissue mass index (LSTMI), not in children with high LSTMI.

Ulbricht et al. [146], in a study on adolescents in South Brazil, did not find a direct relationship between excess body fat and BMD. However, in boys the lowest BMD values were for those presenting two aggregated factors (physical inactivity and sedentary lifestyle), habits considered harmful to bone health by increasing the risk of developing osteopenia/osteoporosis in more advanced ages. Han, Kim, and Kim [150] evaluated the factors related to BMD in adolescents and verified that individuals with lower body fat mass indices, more skeletal muscle, higher BMI, and a higher intake of calcium supplements had higher BMD.

Given the scarcity of evidence relating obesity to bone health in adolescents, new studies are needed to evaluate the effect of overweight/obesity on bone health over more extended periods (infancy and adolescence).

### 4.3. Bone Health and Puberty

Acquisition of bone mass over a lifetime is influenced by several factors, such as genetics, sex, race, nutrition, physical activity, hormone metabolism [136,151], body weight, and diet (food consumption of calcium and vitamin D or supplementation) [152]. The bone mass acquired during the growth spurt peak predicts the BMD of the adult since about 90% of this mass is gathered in that period [46]. Bone build-up occurs progressively from birth to infancy; however, its most expressive acceleration occurs at puberty under the influence of anabolic hormones, such as growth hormone (GH), growth factor linked to insulin (IGF-1), and insulin [153].

Excess body weight can accelerate sexual maturity, which is reflected in accelerated bone age, mainly in obese children and adolescents. Sopher et al. [154], assessing bone age in prepubertal children and premature adrenarche, identified obesity as highly associated with advanced bone age. Silva et al. [155] also verified a relationship between BMI and bone age, especially in overweight girls, who had a higher mean of bone age than those of low or adequate weight; no significant difference was detected in boys. However, Busch et al. [156] investigated the association between obesity and onset of puberty in boys and concluded that a cohort of obese boys starts puberty significantly earlier than the cohort of normal-weight boys.

Furthermore, children with excess weight tend to have a stature below their potential according to the mean parental height [157]. Klein, Newfield, and Hassink [158] and Oh et al. [159] obtained similar results once both studies found an association between advanced bone age and child obesity. The latter identified the prevalence of advanced bone age progression with the severity of obesity. Another interesting result was a higher prevalence of advanced bone age in those with MS, a more severe degree of non-alcoholic hepatic steatosis, and arterial hypertension.

Findings have been reported about the role of epigenetic factors in the physiopathology of bone diseases, such as osteoporosis. Epigenetics is the interaction between genetics and the environment, which produces individualcharacteristics, a phenotype [160]. Such epigenetic changes do not interfere with DNA structure but can alter the function of the genes that produce specific phenotypical characteristics [161]. Experimental studies have pointed to specific microRNAs in the serum of post-fracture women and their link with bone health [162,163]. MicroRNAs are small RNAs that regulate gene expression [164]. They give distinct cellular responses in both bone formation [165] and bone disease [163], which indicates the possibility of a diagnosis and prognosis for bone diseases and susceptibility to fracture [162].

Bone development in the intrauterine environment has received prominence in research into the pathology of osteoporosis. The maternal nutritional status and exposure to stress during pregnancy can affect the epigenetic status of several genes during fetal bone development and hinder them. Furthermore, the epigenetic changes can be transmitted through the mother or father and can even affect future generations [166]. A Danish study of a birth cohort found that a poor diet during pregnancy was associated with a higher risk of fractures in infancy [167], confirming that the maternal diet and intrauterine environment influences infant bone health.

## 5. Conclusions

Most studies point to the negative consequences of excess weight—mainly visceral adiposity—on bone health and fragility, which is linked to the production of pro-inflammatory cytokines, increased bone reabsorption, and damage to bone formation.

It is recommended that a biochemical evaluation and an assessment of blood pressure at the pubertal stage be included in the routine health check-ups of children and adolescents, especially those with excess body fat. The aim is to identify the level of sexual maturity and possible early bone and metabolic commitment with obesity in addition to an evaluation of food consumption and life habits.

Another relevant aspect is checking vitamin D serum levels, since hypovitaminosis D is a public health problem that negatively affects metabolic parameters such as bone health. In general, most adolescents are sedentary and do not engage inoutdoor physical activity. Indeed, cutaneous synthesis is a relevant contributor for adequate vitamin D levels; thus, it is recommended that adolescents be stimulated to engage inphysical activity in the open.

Promoting healthy habits is crucial for promoting health, especially healthy eating and lifestyles, and for preventing disease, such as osteopenia/osteoporosis. Furthermore, food vigilance and nutrition during the gestational period and lactation should beimproved since epigenetic modifications affect the health of descendants.

## Figures and Tables

**Figure 1 nutrients-14-03260-f001:**
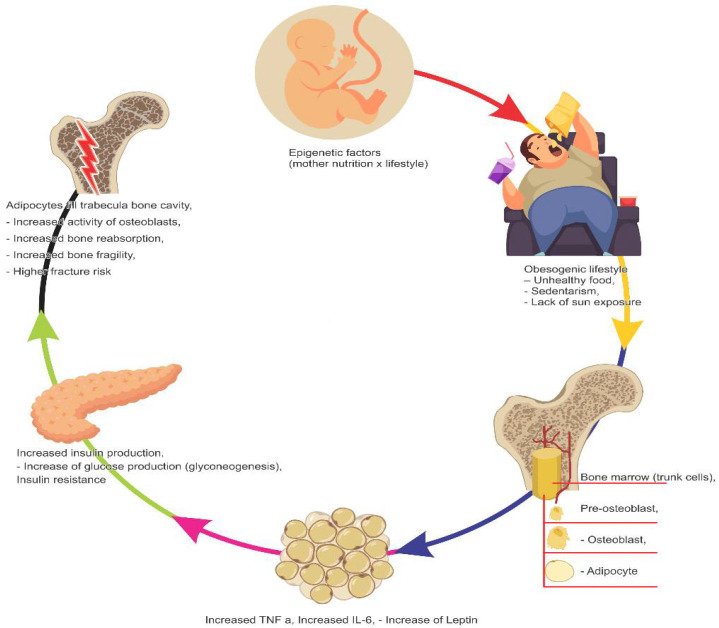
Factors that impair bone health.

## Data Availability

Not applicable.

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
