# Peer review of "Adiposity Metabolic Consequences for Adolescent Bone Health"

_nutrients, 2022, doi:10.3390/nu14163260_

Round 1

Reviewer 1 Report

The proposed review entitled "Adiposity Metabolic Consequences for Adolescent Bone Health" clearly describes the interaction and possible underlying mechanisms linked to bone health under obese condition. It includes the main players in bone metabolism. The proposed manuscript highlights changes during growth and the particular contribution of overweight/obesity on bone health in later life. This manuscript pinpoints to the necessity to research in this field. 

The manuscript is well written and interesting to the researchers in this fields by contributing with an overview about knowns and unknowns.

Some minor comments need to be addressed:

Line 244: please use the term 30-35%

Line 455: There is a mistake in the title: it should be "MS" instead of "SM"

Line 459: You are writing about vitamin D and use the term "vitamin" as a description. To be scientifically clear, please describe vitamin D as a hormone.

Page 11: The Figure lacks a figure number and description.

Author Response

Dear Reviewer 1,

We are submitting the manuscript titled “Adiposity metabolic consequences for adolescent bone health” for possible publication in the Nutrients. The review article was submitted a Section Nutrition and Public Health, and Special Issue entitled "Nutrition and Bone Health", to be published in the Nutrients journal (ISSN 2072-6643, IF 6.706), a special issue that fits perfectly with our theme.

We thank you for your considerations in our manuscript with the purpose of improving our article. We analyzed and modified our article according your considerations and the changes are listed below. In the manuscript, these changes are highlighted in yellow.

Line 244: please use the term 30-35%

Reply: Done

Line 455: There is a mistake in the title: it should be "MS" instead of "SM"

Reply: Done – new line - 457

Line 459: You are writing about vitamin D and use the term "vitamin" as a description. To be scientifically clear, please describe vitamin D as a hormone.

Reply: We insert a new paragraph with this approach between the 467 and 473 lines. The new references are 123 and 124 (pages 19 and 20).

Page 11: The Figure lacks a figure number and description.

Reply: Done. Line 532

Thank you for your consideration,                                                      

Best regards,

Karine de Cássia Freitas.

Reviewer 2 Report

The review is very adequate and complete. The discussion is pertinent and the conclusions agree with the data provided.

Author Response

Dear Reviewer 2,

We are submitting the manuscript titled “Adiposity metabolic consequences for adolescent bone health” for possible publication in the Nutrients. The review article was submitted a Section Nutrition and Public Health, and Special Issue entitled "Nutrition and Bone Health", to be published in the Nutrients journal (ISSN 2072-6643, IF 6.706), a special issue that fits perfectly with our theme.

First, we really thank you for all Comments and Suggestions, as we know that reading and analyzing the scientific article takes time and dedication. We hope that this review may alert for the risk that obesity (specifically, accumulated visceral fat) may harm bones in the infant–juvenile phase, thereby increasing osteopenia/osteoporosis in adults and the elderly.

Best regards,

Karine de Cássia Freitas.
